# Unveiling a Health Disparity: Comparative Analysis of Head and Neck Cancer Trends between First Nations People and Non-Indigenous Australians (1998–2015)

**DOI:** 10.3390/cancers16142548

**Published:** 2024-07-15

**Authors:** Lamia Fahad Khan, Santosh Kumar Tadakamadla, Jyothi Tadakamadla

**Affiliations:** 1Dentistry and Oral Health, Department of Rural Clinical Science, La Trobe Rural Health School, La Trobe University, Bendigo, VIC 3550, Australia; s.tadakamadla@latrobe.edu.au (S.K.T.); j.tadakamadla@latrobe.edu.au (J.T.); 2Violet Vines Marshman Centre for Rural Health Research, La Trobe Rural Health School, La Trobe University, Bendigo, VIC 3550, Australia

**Keywords:** First Nations people, head and neck cancer, incidence, mortality, survival

## Abstract

**Simple Summary:**

This study focused on the trends of Head and Neck Cancers (HNC) in the First Nations people of Australia. There is an underwhelming amount of literature holistically analysing the trend of HNC among the First Nations people. This study comprehensively analysed incidence, mortality, and survival rates in the First Nations people, and compared these trends with the non-Indigenous Australian population. This emphasized the need to investigate the underlying causes and barriers for differences in the HNC burden between First Nations people and non-Indigenous Australians. Collaborative efforts, spanning from local to national levels, are required to address the HNC burden in Australia, particularly, in First Nations communities.

**Abstract:**

Background: We aim to assess and compare the HNC trends between the First Nations and non-Indigenous population. Methods: HNC incidence (1998–2013) and mortality (1998–2015) data in First Nations people and non-Indigenous Australians were utilised from the Australian Cancer Database. The age-standardised incidence and mortality trends along with annual percentage changes were analysed using Joinpoint models. Age-standardised incidence and mortality rates according to remoteness, states, and five-year survival rates among First Nations people and non-Indigenous Australians were presented as graphs. Results: First Nations people had over twice the age-standardised incidence (2013; 29.8/100,000 vs. 14.7/100,000) and over 3.5 times the age-standardised mortality rates (2015; 14.2/100,000 vs. 4.1/100,000) than their non-Indigenous counterparts. Both populations saw a decline in mortality, but the decline was only statistically significant in non-Indigenous Australians (17.1% decline, 1998: 4.8/100,000, 2015: 4.1/100,000; *p* < 0.05). Across all remoteness levels and states, First Nations people consistently had higher age-standardised incidence and mortality rates. Furthermore, the five-year survival rate was lower by 25% in First Nations people. Conclusion: First Nations people continue to shoulder a disproportionate HNC burden compared to non-Indigenous Australians.

## 1. Introduction

Head and neck cancers (HNC) encompasses a diverse group of malignancies affecting the oral cavity including the sinuses, salivary glands, nasopharynx, oropharynx, and the larynx [1]. In Australia, the incidence of HNC has shown a concerning upward trajectory, comprising an estimated 3.2% of all cancer diagnoses in 2022, and 2.5% of all cancer-related deaths in the same year [2]. It is also the seventh most commonly occurring cancer worldwide [3]. Recognised risk factors for HNC include tobacco consumption, excessive alcohol intake, and infection with certain types of human papillomavirus (HPV) [1].

Aboriginal and Torres Strait Islanders (respectfully referred to here onwards as First Nations people) comprise 3.3% of the total Australian population and have historically faced a heightened susceptibility to cancer [4]. This vulnerability extends to HNC with First Nations people exhibiting at least a two-fold higher incidence and mortality rate compared to non-Indigenous Australians [5]. The average 1-year and 5-year survival rate of HNC is also lower in First Nations people compared to non-Indigenous Australians [5]. This disparity is further accentuated within the First Nations people, where men are at least twice as likely to be diagnosed with and succumb to HNC compared to women, often a reflection of modifiable risk factors in males such as smoking and alcohol consumption [5]. Consequently, First Nations people suffer a health burden from HNC disproportionate to non-Indigenous Australians [6].

The existing body of literature pertaining to HNC within the Australian context is comprehensive in some regards but notably lacking in others. Certain studies focus on the whole Australian population [7,8,9,10], whereas others delve into specific facets of the issue such as HPV-associated HNC [11,12,13,14] and state-based trend analysis [15,16,17,18]. The existing literature lacks a holistic approach considering HNC incidence, mortality, and survivorship, but more crucially, there is an underwhelming amount of research focusing on First Nations people. This omission is significant given the unique challenges and health disparities experienced by First Nations communities in Australia [6]. A comprehensive understanding of HNC, coupled with a dedicated focus on First Nations people, is essential for shaping effective healthcare policies, interventions, and support systems [6]. Therefore, the objective of this research is to present the trend of HNC incidence and mortality among First Nations people and draw comparisons with non-Indigenous Australians by retrospectively assessing the data available from the Australian Cancer Database (ACD). In addition, we intend to explore the incidence and mortality rates among First Nations people and non-Indigenous Australians across the regions based on remoteness and states.

## 2. Materials and Methods

The dataset for this study was obtained from the Australian Cancer Database (ACD), the authoritative source for comprehensive cancer-related data in Australia [19]. Legislation mandates the reporting of all new cancer cases to the central cancer registry. This data is annually submitted to the Australian Institute of Health and Welfare (AIHW) through the Australasian Association of Cancer Registries, which then compiles and oversees the ACD [19]. HNC cases in the dataset adhere to anatomical site classifications defined by International Classification of Diseases, 10th Revision (ICD-10) codes [20]. Specifically, HNC encompasses neoplastic cases falling under ICD-10 codes C00–C14 and C30–32, covering malignant neoplasms of the lip, oral cavity, pharynx, nasal cavity, paranasal sinuses, and larynx [1].

The primary data table for this study was derived from the folder “Cancer in Aboriginal and Torres Strait Islander People of Australia,” which was narrowed down to specifically focus on HNC. This dataset provided comprehensive information on HNC incidence and mortality across First Nations statuses, years, sexes, ages, states, levels of remoteness, and 1-year and 5-year relative survival rate. According to the ACD, relative survival rate refers to the probability of being alive for a given amount of time after cancer diagnosis, compared to the expected survival rate of a comparable group in the Australian population adjusted for age, sex, and calendar year [21]. The most recent data in the ACD was available from 1998 to 2015, and the data analysed in this study were incidence and mortality categorised by First Nations status, sex, remoteness, and state distribution, and the 5-year relative survival rate. Sex-specific and population level incidence and mortality data were available in the ACD from 1998 to 2013 and 1998 to 2015, respectively. Data pertaining to remoteness and state distribution were available for the following time periods: 2009–2013 for incidence and 2011–2015 for mortality. Five-year relative survival percentages were available for two specified periods: 1999–2006 and 2007–2014. It should be noted that no annual level data were available in the ACD for incidence and mortality categorised by remoteness, state, or the n-year relative survival rate. Additionally, no person-level data was available for HNC incidence, mortality, or survivorship. The ACD employs the Australian Standard Geographical Classification- Remoteness Area (ASGC-RA) classification to assess remoteness levels in Australia, categorising areas into groups such as ‘major cities,’ ‘inner regional,’ ‘outer regional,’ ‘remote,’ and ‘very remote’ [22].

### Statistical Analysis

The ACD provides age-standardised incidence and mortality rates per 100,000 persons categorised by sex, remoteness, state distribution, and First Nations status derived using the direct-age standardisation method utilising the 2001 Australian standard population. Joinpoint regression analysis was employed to analyse the trend and compute annual percent changes (APCs) for sex-specific incidence and mortality rates for both First Nations people and non-Indigenous Australians. Joinpoint regression uses log transformation of the incidence or mortality rates resulting in a linear model fitted to the trend on a logarithmic scale facilitating the direct comparison of relative differences and annual percent changes over a specific period in the output [23]. This method also aims to identify significant changes in rates over time utilising a t-test to compare the slope changes, defined at *p* < 0.05 [23]. A non-significant change in rates indicated a stable trend. Analysis was undertaken using ‘Joinpoint Trend Analysis Software v4.9’ from the National Cancer Institute in the USA [24]. Joinpoint regression analysis remains the gold standard for assessment of time-trend cancer data [23].

Descriptive data on age-standardised incidence and mortality rates according to remoteness and states in Australia from the ACD were used to generate graphs to facilitate visual comparison between the First Nations people and non-Indigenous Australians. The ACD did not publish data pertaining to age-standardised incidence rates in South Australia (SA), and age-standardised mortality rates in Victoria (VIC) in both First Nations and non-Indigenous populations. Also, the ACD had no HNC incidence and mortality data from Tasmania (TAS) and the Australian Capital Territory (ACT).

Five-year relative survival rates were reported by the ACD as specific percentages for two distinct time periods (1999–2006 and 2007–2014) among First Nations and non-Indigenous Australians. Graphs were generated using GraphPad Prism v8.4. Certain data in the ACD datasets were given as n.p. meaning not publishable because of small numbers, confidentiality, or other concerns about the quality of the data. Such data were very minimal and data boxes were left empty when creating the graphs.

## 3. Results

### 3.1. Incidence

The overall incidence of HNC in First Nations people is higher than non-Indigenous Australians. Between the period 1998–2013, the incidence amongst First Nations females (Figure 1A) was at least 1.9 times higher than their non-Indigenous counterparts (Figure 1B) (2013; 14.0/100,000 vs. 7.4/100,000), and approximately 2.2 times higher for First Nations males (Figure 1C,D) (2013; 48.0/100,000 vs. 21.8/100,000). First Nations males have at least 3.4 times higher incidence than First Nations females, and non-Indigenous males have at least 2.9 times higher incidence than non-Indigenous females. Joinpoint regression analysis demonstrated that incidence rates for HNC have remained stable (*p* > 0.05) over 1998–2013 for all First Nations and non-Indigenous populations except for non-Indigenous males. There has been a decrease in overall incidence in HNC for non-Indigenous males with an annual percentage change (APC) of −2.68% (*p* < 0.05) between 1998 and 2003, and −1.16% (*p* < 0.05) between 2006 and 2013 (Figure 1D). Over the same period, the incidence in First Nations males has increased (Figure 1C), although this was not statistically significant (APC = 1.41, *p* > 0.05). Population-level incidence rates for the First Nations and non-Indigenous population (Figure 2) demonstrated stability; whilst overall incidence has increased in the First Nations population (APC = 0.33, *p* > 0.05) and decreased in the non-Indigenous population (APC = −0.01 to −2.81, *p* > 0.05), neither trend was statistically significant. Notably, between 1998 and 2013, First Nations people demonstrated higher incidence rates compared to non-Indigenous Australians (27.3–27.8/100,000 vs. 14.7–16/100,000), and the same trend applied to males and females in both populations. Overall, incidence amongst First Nations people was at least 2 times higher than their non-Indigenous counterparts (2013; 29.8/100,000 vs. 14.7/100,000).

### 3.2. Mortality

Compared to their non-Indigenous counterparts, First Nations female (Figure 3A) mortality was at least 6.8 times higher (2015; 11.5/100,000 vs. 1.7/100,000) and male mortality was at least 3.4 times higher (2015; 22.0/100,000 vs. 6.5/100,000) (Figure 3B–D). In both populations, males have higher HNC mortality rates than females. First Nations males experienced at least 2-fold higher mortality, and non-Indigenous males at least 3.8-fold higher mortality than their female counterparts. Joinpoint regression analysis demonstrated overall declines in mortality for both First Nations and non-Indigenous males, with decreases of 13.6% (1998: 25/100,000, 2015: 22/100,000) and 20% (1998: 7.8, 2015: 6.5), respectively. However, neither of these declines were statistically significant (*p* > 0.05) with an APC of −0.83 between 1998 and 2015 in First Nations males, and APCs of 2.12 (1998–2001), 6.72 (2001–2004), and −0.43 (2004–2015) in non-Indigenous males. Non-Indigenous females were the only population subset to demonstrate a statistically significant decline in mortality with an overall 25.7% decrease (1998: 2.2/100,000, 2015: 1.75/100,000), and an APC of −1.46 (*p* < 0.05) from 1998 to 2015. Conversely, First Nations females demonstrated an upward trend in mortality with a 28.7% increase over the years (1998: 8.2/100,000, 2015: 11.5/100,000) with an APC of 1.99 (*p* > 0.05); however, this was not statistically significant. At the population level, both First Nations (Figure 4A) and non-Indigenous (Figure 4B) populations had overall decreases in mortality with declines of 10.6% (1998: 15.7/100,000, 2015: 14.2/100,000) and 17.1% (1998: 4.8/100,000, 2015: 4.1/100,000), respectively. There was an APC of −0.59 (*p* > 0.05) between 1998 and 2015 in First Nations people, and APCs of 1.43 (1998–2001) and 0.24 (2006–2015) in non-Indigenous Australians, with none of these trends being statistically significant (*p* > 0.05). The only statistically significant trend was a decline in mortality in non-Indigenous Australians between 2001 and 2006 with an APC of −4.55 (*p* < 0.05). Again, from 1998 to 2015, First Nations people consistently had higher mortality rates compared to non-Indigenous Australians (14.2/100,000–15.7/100,000 vs. 4.1/100,000–4.8/100,000), and the same trend applied to males and females in both populations. Overall, First Nations people had at least 3.5 times higher mortality than their non-Indigenous counterparts (2015; 14.2/100,000 vs. 4.1/100,000).

### 3.3. Remoteness

At each level of remoteness, HNC incidence (Figure 5A) and mortality rates (Figure 5B) were higher in First Nations people compared to non-Indigenous Australians, with the disparity worsening as remoteness increases, especially regarding mortality. This trend is evident, with a gradual rise in both HNC incidence and mortality rates as remoteness increases, peaking in very remote areas. For instance, First Nations people in very remote areas (2009–2013) had 1.9 times higher HNC incidence (39.3/100,000 vs. 20.6/100,000) and four times higher mortality (2011–2015) (23.2/100,000 vs. 5.8/100,000) compared to non-Indigenous Australians. To accentuate the impact of rurality in widening the disparity in incidence and mortality rates, First Nations people in very remote areas had 2.1 times higher incidence (39.3/100,000 vs. 18.9/100,000) and 3.6 times higher mortality (23.1/100,000 vs. 6.5/100,000) than their counterparts residing in major cities. For First Nations people, the age-standardised incidence rate for HNC ranged from 19.0/100,000 to 39.3/100,000, compared to a range from 12.8/100,000 to 20.6/100,000 in non-Indigenous Australians. Similarly, for First Nations people, the age-standardised mortality rate for HNC ranged from 6.4/100,000 to 23.1/100,000, compared to 3.2/100,000 to 5.8/100,000 in non-Indigenous Australians.

### 3.4. States

Across the Australian states examined, First Nations people consistently exhibited higher HNC incidence and mortality rates compared to their non-Indigenous counterparts (Figure 6A,B). Northern Territory (NT) recorded the highest incidence rates (2009–2013) and mortality rates (2011–2015) across both populations, with First Nations people experiencing a 1.4 times higher incidence rate (43.0/100,000 vs. 30.5/100,000) and a 3.2 times higher mortality rate (28.9/100,000 vs. 8.9/100,000) compared to non-Indigenous Australians in the NT. Conversely, VIC reported the lowest incidence rates for both First Nations people and non-Indigenous Australians, although First Nations people still had a 1.5 times higher incidence rate than their non-Indigenous counterparts (18.7/100,000 vs. 12.3/100,000). The most notable disparity in HNC incidence was observed in Western Australia, with First Nations people having a 2.2 times higher incidence rate than non-Indigenous Australians. Regarding mortality rates, the NT exhibited the most substantial disparity between the First Nations and non-Indigenous populations. Queensland reported the lowest mortality rate for First Nations people (7.6/100,000), while SA held this distinction for non-Indigenous Australians (3.4/100,000).

### 3.5. Survival

First Nations people exhibited a lower 5-year relative survival rate after being diagnosed with HNC compared to non-Indigenous Australians (Figure 7). From 1999 to 2006, First Nations people had a 24.7% lower chance of survival (36.7% vs. 61.4%), and between 2007 and 2014, First Nations people had a 24.5% less chance of survival (41.5% vs. 66%). There has been an elevation of 4.8% and 4.6% in 5-year survival rates in First Nations (1999–2006: 36.7%, 2007–2014: 41.5%) and non-Indigenous (1999–2006: 61.4%, 2007–2014: 66%) populations over the two distinct time intervals, respectively.

## 4. Discussion

Our study revealed higher incidence and mortality rates among First Nations people than non-Indigenous Australians over the last 17 years (1998–2015); there has been no improvement in HNC incidence or mortality amongst the First Nations population. In addition, the burden of HNC was exacerbated by level of remoteness, and geographical location. First Nations people not only bear a significantly higher burden of HNC, but lower survival rates as well. First Nations people consistently experience an incidence rate at least double that of non-Indigenous Australians, and a mortality rate at least 3.5 times higher as well. Equally concerning is the disparity in gender, with males in both populations more susceptible to HNC.

On the other hand, there were statistically significant declines in incidence and mortality in the non-Indigenous cohort. More specifically, there was a decrease in incidence in non-Indigenous males with an APC of −2.68 (*p* < 0.05) between 1998 and 2003, and −1.16 (*p* < 0.05) between 2006 and 2013, and a decline in mortality in non-Indigenous females between 1998 and 2015 with an APC of −1.46 (*p* < 0.05). At the population level, there was a decline in mortality of non-Indigenous Australians between 2001 and 2006 with an APC of −4.55 (*p* < 0.05). These significant declines are encouraging findings in the non-Indigenous cohort; however, more targeted interventions are required to produce such results consistently at the population-level.

To comprehend the underlying causes behind the rising incidence and mortality rates of HNC in the First Nations population, it is imperative to delve into the potential risk factors that contribute to this heightened risk. Solar radiation is a risk factor for HNC, typically linked to lip cancer and disproportionately affects non-Indigenous Australians [25]. However, tobacco and alcohol consumption, inadequate nutrition, and HPV infection could be the risk factors leading to the high incidence of HNC in First Nations people [26,27,28,29].

Tobacco and alcohol are significant public health concerns for First Nations people [26]. In 2012–2013, First Nations people were 2.6 times more likely than non-Indigenous Australians to be current daily smokers (40% vs. 15%), often commencing at a younger age [27]. Additionally, while First Nations people are less likely to consume alcohol than non-Indigenous Australians, those who do drink do so at hazardous levels with one fifth drinking at levels exceeding national alcoholism risk guidelines [28]. Rates of heavy and binge drinking are notably elevated in rural and remote First Nations communities [29], aligning with our findings of higher HNC incidence and mortality in these regions. While this paper could not quantify risk factors and correlate them with incidence rates, it is probable that tobacco and alcohol use in the First Nations community are linked to their higher HNC incidence and mortality rates, especially among males.

Poor nutrition also contributes to the development of HNC, as fruits and vegetables are rich in antioxidants and minerals crucial for dental health. In contrast, processed foods lack essential nutrients, leading to oxidative stress and DNA damage [30]. Overall, First Nations people have poorer nutrition status than non-Indigenous Australians, attributable to limited access to healthy and affordable food, especially in remote communities [31]. With a diet typically high in energy and sugars, First Nations people are at increased risk of HNC [31].

Another significant risk factor for HNC is HPV infection with strains HPV-16 and HPV-18 strongly implicated in HNC [11,32,33]. This relationship was established because rising HPV cases in Australia was concomitant with rising HPV-related HNC [12,13]. A study revealed that the prevalence of oral HPV infection was approximately 15 times higher in First Nations people compared to the general population [14]. A pivotal moment in Australia’s efforts to combat HPV-associated diseases was the introduction of a vaccination program in 2007 [12]. However, a distinctive outcome of this effort, which primarily targets female vaccination, is that while cervical cancer cases have declined, HPV-related HNC cases continue to surge [12]. This phenomenon has raised concerns regarding the disproportionate burden of HNC borne by males in comparison to females [13]. Therefore, the impact of HPV vaccination on rising HNC cases remains to be fully realised [11]. In addition, despite the wide availability of the quadrivalent HPV vaccine in Australia, First Nations communities have lower HPV vaccination rates and completion rates compared to non-Indigenous communities [33]. There is no national-level data available for the uptake of HPV vaccination among First Nations people [34]. The lower vaccination rates may stem from historical government-mediated discrimination, which could erode trust in the healthcare system and foster anti-vaccination cultural beliefs, such as viewing it as the ‘white man’s medicine’ [35]. This presents a significant health challenge for First Nations people [14], considering their higher HNC incidence and mortality rates.

Higher levels of incidence and mortality rates of HNC in First Nations people compared to non-Indigenous Australians could also be the outcome of a multifaceted interplay of barriers encompassing affordability, access, and socioeconomic disadvantage. Socio-economic disadvantage serves as a pivotal driver in the stark health disparities existing between First Nations and non-Indigenous Australians [36]. This disadvantage is deeply rooted in historical dislocation and trauma of First Nations people, compounded by systemic discrimination across education, employment, and housing [37]. As a result of having a lower median household income than the Australian population, more than two in five First Nations people aged over 15 defer or entirely avoid dental care due to the associated costs of consultations, medications, and transportation [36]. For the First Nations people who do seek public dental care, long waiting periods and stringent appointment scheduling further limit their access to dental services, leading to delays in the diagnosis and treatment of HNC [36]. These delays are mirrored in higher mortality rates and lower survival rates among First Nations people [16]. Regrettably, unlike cervical and breast cancer, Australia lacks a regular screening tool for HNC [38]. Consequently, the diagnosis of HNC is primarily opportunistic or incidental, often occurring during routine dental visits [38]. Also, postponement of dental visits may result in many cases of HNC remaining undiagnosed, potentially leading to an underestimation of the actual prevalence of HNC in both populations, with particular implications for the First Nations community [39].

First Nations people encounter cultural barriers that could also function as deterrents to seeking dental care, consequently elevating the risk of late-stage cancer detection, higher mortality rates, and reduced survival prospects. Firstly, the limited cultural representation in the oral health workforce can result in experiences of discrimination and issues with decision-making [40]. Secondly, many dental services lack cultural sensitivity and maintain strict appointment schedules, potentially deterring First Nations people who prefer to visit dentists with family and friends [36].

Residents of rural and remote areas face challenges in accessing dental care due to geographic isolation, limited infrastructure, and inadequate funding, which often means that dental services are supplemented by models such as a fly-in-fly-out service [36]. Australians living in regional and remote areas often have substandard access to community-controlled health organisations. There is also a high density of First Nations people in these areas, with over 43% of First Nations people living in regional Australia and 21% in remote areas [36]. Population-based studies indicate that First Nations people in regions with available dental services tend to underutilise these services, attributable to the socio-economic and cultural factors outlined above [35]. In 2021–2022, Australians living in major cities (51%) were more likely to have seen a dental professional than those living in inner regional areas (45%) or outer regional, remote, and very remote areas (43%) [36]. Moreover, the dearth of specialised cancer therapy services in rural and remote areas amplifies disparities in oral health outcomes when juxtaposed with metropolitan regions [36]. Given that these cancer services are predominantly situated in major cities, Australians may opt to forgo such treatments, even when medically advised, due to the considerable travel and cost involved [37]. A study exploring the impact of remoteness on people with HNC found that individuals in rural areas were significantly less likely to undergo multidisciplinary treatment reviews, receive treatment, and start treatment within 30 days of diagnosis [41]. This glaring contrast in access to HNC treatment between major cities and very remote areas, coupled with the concentration of First Nations people in these remote regions, mirrors the higher mortality and lower survival rate within the First Nations population [41].

There is a wide variability of HNC incidence and mortality in First Nations and non-Indigenous populations across Australian states. However, the NT emerged with the highest HNC incidence and mortality rates in both populations. The NT houses the largest First Nations population in Australia (32%), with a higher prevalence in remote communities [15]. Tobacco use is prevalent among First Nations individuals in the NT, with nearly half being smokers (2007–2008), compared to 28% in the non-Indigenous population [10]. The NT reports the highest smoking rate among all Australian states [9]. Additionally, per capita alcohol consumption in the NT is approximately 15 litres of pure alcohol annually, significantly exceeding the national average [10]. Furthermore, malnutrition affects at least 20% of the First Nations population in the NT, contributing to chronic diseases and exacerbating health challenges [10]. The convergence of these factors in the NT, coupled with its substantial First Nations population residing in remote areas, underscores the pressing need to enhance early screening and diagnosis of HNC in this region [42].

First Nations people were found to present with 24.5% lower relative 5-year survival rate after being diagnosed with HNC (2007–2014; 41.5% vs. 66%). This difference can be primarily ascribed to the advanced stage at which First Nations people typically receive their HNC diagnoses [43]. Data suggest that First Nations people present with regional spread of HNC at a rate of 89% higher than their non-Indigenous counterparts, and about 240% higher likelihood of exhibiting distant spread, presumably resulting in metastatic disease [44]. These differences highlight the critical need to address healthcare accessibility issues in rural and remote Indigenous communities as a crucial step toward improving overall survival rates [16].

This study provides a comprehensive overview of HNC trends in First Nations people and non-Indigenous populations in Australia from 1998 to 2015. However, it has several limitations. Firstly, the time trend data were only available between 1998 and 2015. However, to the best of the authors’ knowledge, this is the first study to comparatively analyse HNC trends in Australia in relation to First Nations people and non-Indigenous Australians. Also, this study used the population-level descriptive data available from the ACD to conduct a time-trend analysis. Additionally, there were some data classified by the ACD as not publishable; therefore, certain graphs are incomplete or data points incomparable between populations. Secondly, there are no HNC incidence data for SA or mortality data for VIC, and no HNC data for TAS and ACT. This makes it challenging to determine the states with the highest and lowest HNC incidence and mortality rates across Australia. Furthermore, no information is provided on the anatomical locations of HNC and their corresponding incidence and mortality rates in this study. It is crucial to note that this study relies on a national database review, which in turn relies on the granularity and accuracy of ACD and AIHW results. Given that these databases are not patient registries, the study was unable to link diagnoses to individual patients. Consequently, the assessment of additional demographic information, risk factors, staging, treatment, or survival outcomes was not feasible. This limitation restricts our capacity to gain deeper insights into the state of HNC in Australia.

## 5. Conclusions

This study found that First Nations people presented with higher incidence and mortality rates, and lower 5-year survival rates, than their non-Indigenous counterparts between 1998 and 2015. In addition, the HNC incidence and mortality rates differed between states, and according to the remoteness levels. Our study underscored the urgent need for a comprehensive understanding of HNC in the First Nations community, recognising the complex interplay of health, socio-economic factors, and cultural determinants. To gain deeper insights into HNC epidemiological trends, aetiology, and prognosis among Australians, further longitudinal studies are essential.

## Figures and Tables

**Figure 1 cancers-16-02548-f001:**
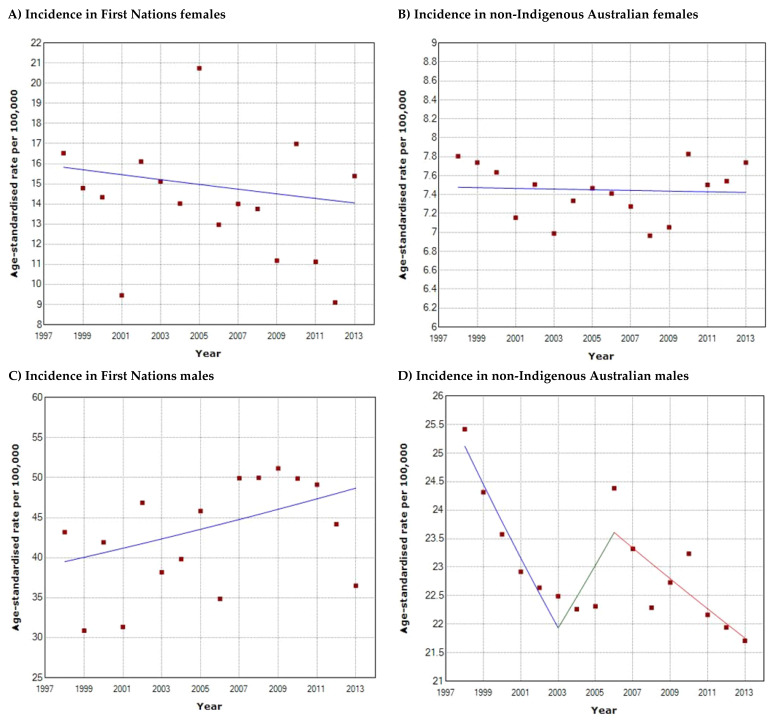
Head and neck cancer incidence trends from 1998 to 2013 in (**A**) First Nations females; (**B**) non-Indigenous Australian females; (**C**) First Nations males; (**D**) non-Indigenous Australian males. The *x*-axis denotes the calendar years from 1998 to 2013 and the *y*-axis denotes the age-standardised incidence rate per 100,000 persons.

**Figure 2 cancers-16-02548-f002:**
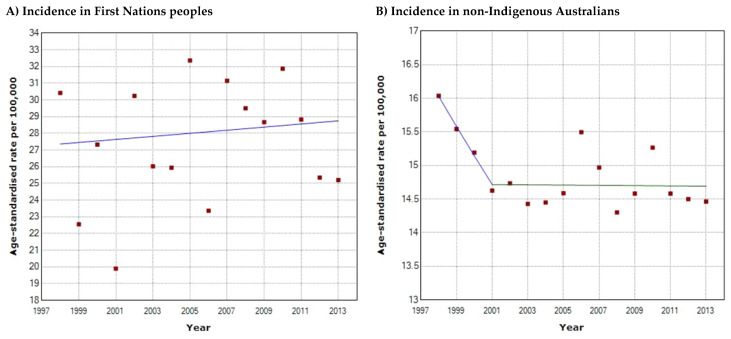
Head and neck cancer incidence trends from 1998 to 2013 in (**A**) First Nations people; (**B**) non-Indigenous Australians. The *x*-axis denotes the calendar years from 1998 to 2013 and the *y*-axis denotes the age-standardised incidence rate per 100,000 persons.

**Figure 3 cancers-16-02548-f003:**
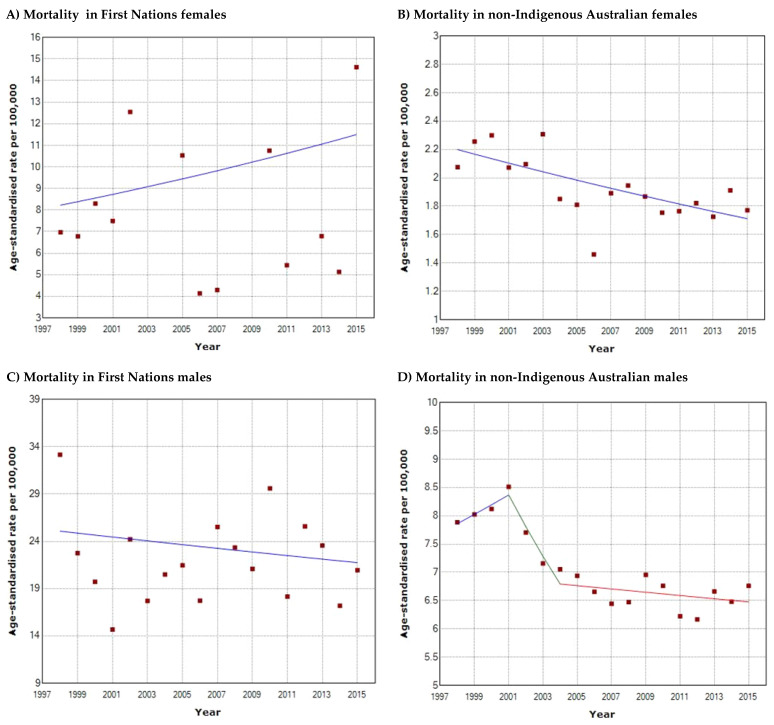
Head and neck cancer mortality trends from 1998 to 2015 in (**A**) First Nations females; (**B**) non-Indigenous Australian females; (**C**) First Nations males; (**D**) non-Indigenous Australian males. The *x*-axis denotes the calendar years from 1998 to 2015 and the *y*-axis denotes the age-standardised mortality rate per 100,000 persons.

**Figure 4 cancers-16-02548-f004:**
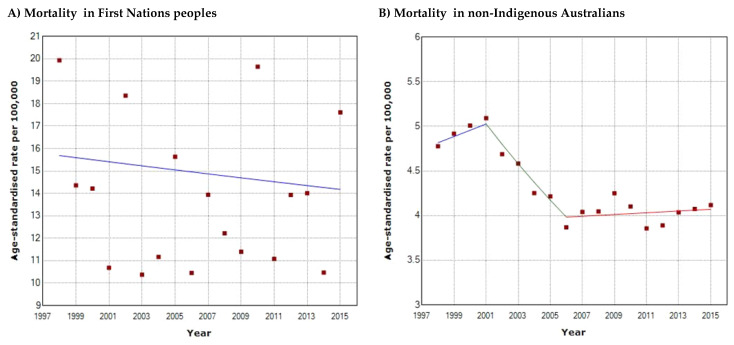
Head and neck cancer mortality trends from 1998 to 2015 in (**A**) First Nations people; (**B**) non-Indigenous Australians. The *x*-axis denotes the calendar years from 1998 to 2015 and the *y*-axis denotes the age-standardised mortality rate per 100,000 persons.

**Figure 5 cancers-16-02548-f005:**
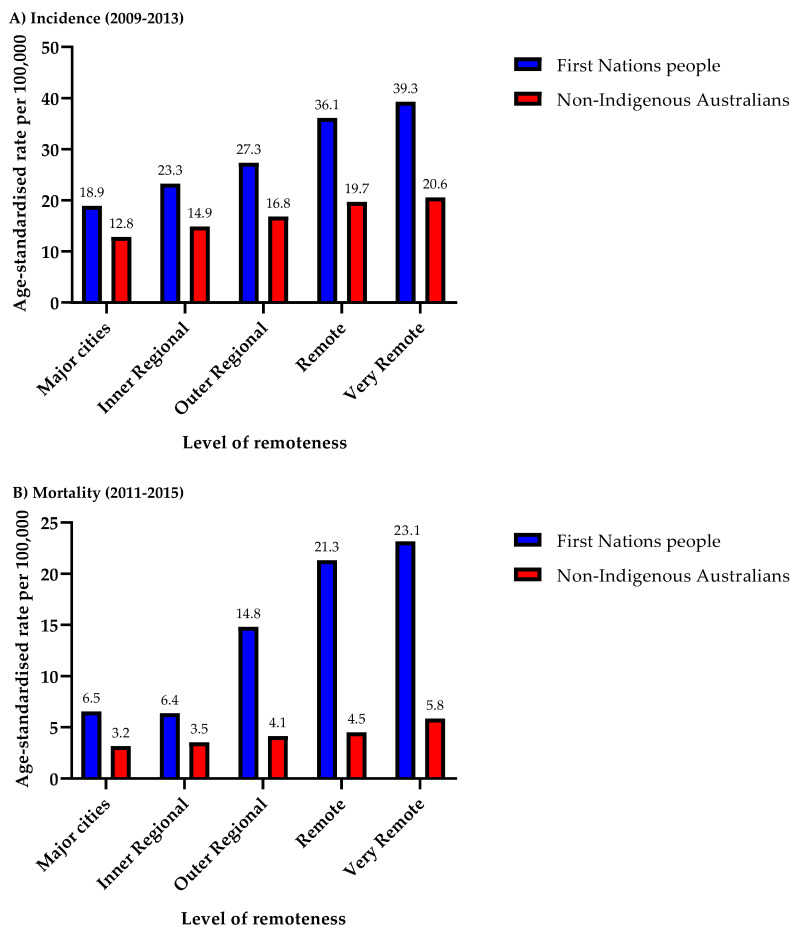
Head and neck cancer incidence and mortality rates by remoteness for First Nations people and non-Indigenous Australians: (**A**) 2009–2013 for incidence rates, and (**B**) 2011–2015 for mortality rates. The *x*-axis denotes the levels of remoteness, and the *y*-axis denotes the age-standardised rate per 100,000 persons.

**Figure 6 cancers-16-02548-f006:**
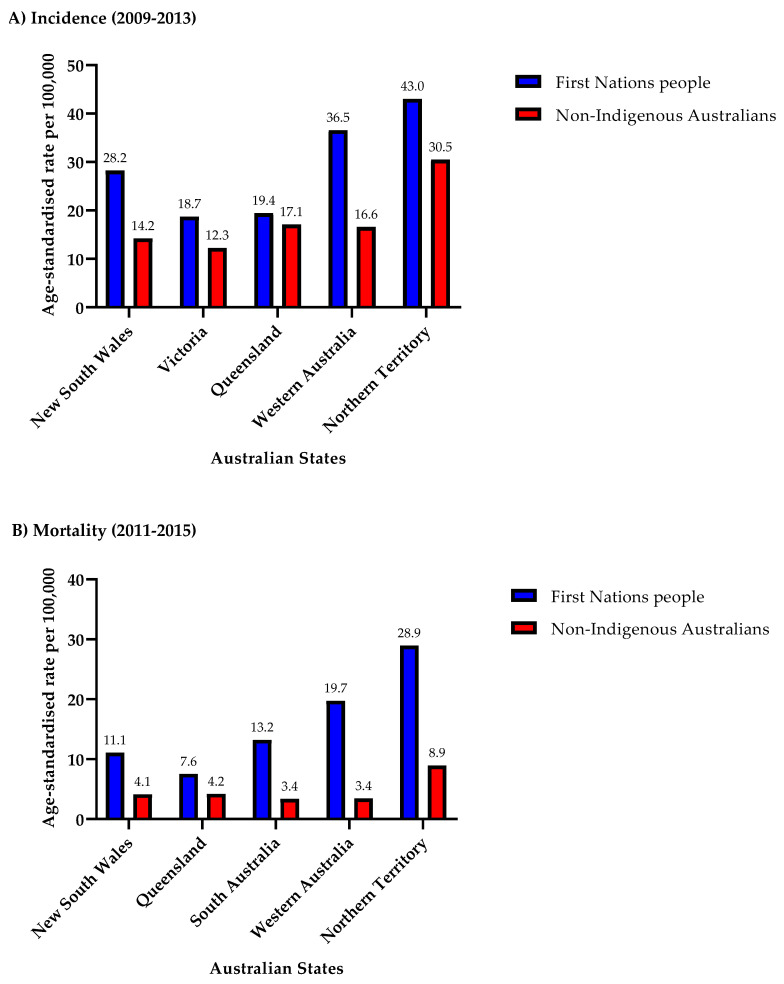
Head and neck cancer incidence and mortality rates by Australian states for First Nations people and non-Indigenous Australians: (**A**) 2009–2013 for incidence rates, and (**B**) 2011–2015 for mortality rates. The *x*-axis denotes the states in Australia analysed in this study and the *y*-axis denotes the age-standardised rate per 100,000 persons.

**Figure 7 cancers-16-02548-f007:**
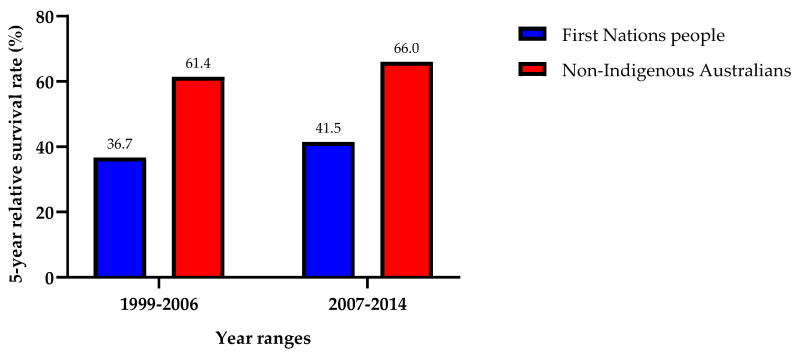
Bar chart comparing the 5-year relative survival (%) of head and neck cancer between First Nations people and non-Indigenous Australians from the time periods 1999–2006 and 2007–2014.

## Data Availability

The data presented in this study are openly available in Australian Cancer Database at https://www.aihw.gov.au/about-our-data/our-data-collections/australian-cancer-database.

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
