# Peer review of "Unveiling a Health Disparity: Comparative Analysis of Head and Neck Cancer Trends between First Nations People and Non-Indigenous Australians (1998–2015)"

_cancers, 2024, doi:10.3390/cancers16142548_

Round 1
Reviewer 1 Report
Comments and Suggestions for Authors
Unveiling a health disparity: comparative analysis of head and neck cancer trends between First Nations peoples and non-Indigenous Australians (1982-2015)
Manuscript Number: cancers-3045830R
Comments to Authors and Editor
The authors focused on the epidemiology of head and neck cancer, with a particular focus on incidence, mortality, and survival rates in the First Nations peoples, and compared these trends with the wider Australian population utilized 1982-2015 data from the Australian Cancer Database. My primary concern relates to the clarity of the methodology, especially in the statistical analysis section. I believe that many readers may struggle to apply the described approach to their own research projects. To enhance the manuscript, it would be beneficial for the authors to comprehensively review and improve the methodology in a scientifically rigorous manner. Specific comments and recommendations are detailed in the following sections.
Rates: There was a lack of a precise definition regarding exposure (population at risk) for calculating the incidence and survival rates. I highly recommend using a well-defined denominator to calculate these rates.
Joint point regression: This regression is valuable for detecting shifts in cancer rates, like sudden increases or decreases in cancer incidence or mortality, possibly attributed to new treatments, public health interventions, or policy changes. However, no changes in incidence rates were observed among First Nations peoples. This regression might be useful for other group, for example, there was a sudden change in incidence rates for non-Indigenous populations.
Poisson/Negative Binomial Regression model: For incidence, I recommend calculating the incidence rates and using a Poisson or Negative Binomial regression model to determine the incidence rate ratio. This will help in deciding whether the incidences significantly differ from the hypothesis-driven approach.
Survival rates: The Kaplan-Meier survival curves are appropriate for determining survival rates based on time-to-event data. This method effectively incorporates censored data to precisely calculate survival rates. The Cox proportional hazard model is suitable for estimating hazard ratios, facilitating comparisons of the risk of death between groups. Alternatively, the Binary logit model can be utilized for estimating 5-year survival rates.
Time frame. The time trend data is very outdated, nearly 10 years old. I strongly urge the authors to use a current data set up to at least 2022 to reflect recent trends. Although the authors covered 33 years of data from 1982 to 2015, most of the analysis is based on data from 1998 onwards.
Appendix and Table 1: Although it was mentioned in the statistical analysis section, I couldn’t find any justification for including Table 1 in the appendix when the data is not available.
Figures. The figures do not follow the professional standards. Please follow the guidelines to ensure the scientific quality of the images.
Author Response
For research article
|
Response to Reviewer 1 Comments
|
||
|
1. Summary |
|
|
|
Thank you very much for taking the time to review this manuscript. Please find the detailed responses below and the corresponding revisions/corrections in track changes in the re-submitted files.
|
||
|
2. Questions for General Evaluation |
Reviewer’s Evaluation |
Response and Revisions |
|
Does the introduction provide sufficient background and include all relevant references? |
Yes |
Please see point-by-point response letter. |
|
Are all the cited references relevant to the research? |
Yes |
|
|
Is the research design appropriate? |
Yes |
|
|
Are the methods adequately described? |
Must be improved |
|
|
Are the results clearly presented? |
Can be improved |
|
|
Are the conclusions supported by the results? |
Can be improved |
|
|
3. Point-by-point response to Comments and Suggestions for Authors |
||
|
Comments 1: Rates: There was a lack of a precise definition regarding exposure (population at risk) for calculating the incidence and survival rates. I highly recommend using a well-defined denominator to calculate these rates.
Response 1: Thank you for pointing this out. We agree with this comment. We want to clarify here that we have not estimated the incidence and survival rates but have instead used the data from the Australian Cancer Database (ACD). As suggested, we have specified in the statistical analysis section the precise denominators used by the ACD for calculating incidence and survival rates. These changes can be found in the manuscript under statistical analysis. |
||
|
Comments 2: Joint point regression: This regression is valuable for detecting shifts in cancer rates, like sudden increases or decreases in cancer incidence or mortality, possibly attributed to new treatments, public health interventions, or policy changes. However, no changes in incidence rates were observed among First Nations peoples. This regression might be useful for other group, for example, there was a sudden change in incidence rates for non-Indigenous populations.
|
||
|
Response 2: Thank you for your comment. We agree that joinpoint regression analysis is a valuable tool for detecting shifts in cancer rates over time and was therefore used in this study aligning with our objective of exploring the trend of Head and Neck Cancer in First Nations and non-Indigenous Australian population. We chose to use the same regression method to analyse incidence and mortality data in First Nations peoples and non-Indigenous Australians to allow direct comparisons and because this was the most appropriate analysis for our dataset. This is discussed further in response to comment 3. Statistically significant trends involving declines in incidence and mortality were seen in non-Indigenous Australians, but such significant changes were not observed among First Nations peoples. The disparity with statistically significant improvements only being observed in the non-Indigenous Australians is of great relevance and is a significant finding of our research.
Comments 3: Poisson/Negative Binomial Regression model: For incidence, I recommend calculating the incidence rates and using a Poisson or Negative Binomial regression model to determine the incidence rate ratio. This will help in deciding whether the incidences significantly differ from the hypothesis-driven approach.
Response 3: Thank you for your comment. We acknowledge that Poisson or Negative Binomial regression analysis would be a useful statistical adjunct. We attempted to explore the possibility of determining the incidence rate ratio and 95% confidence intervals. However, we are very limited by the data available from the ACD and AIHW. The limitations that preclude us from conducting further regression analyses are: 1) the ACD does not provide person-level data for incidence 2) age-related incidence data is available for the period 2009-2013 as a single rate for each age group, hence no annual-level data is available and 3) information regarding age structure further categorised by First Nations status is unavailable to determine the total population according to age, sex and First Nations status. We have also sought consultation from a statistician who viewed our available data and advised us of the above. Furthermore, the strength of our study lies in its presentation of an encompassing population-level dataset. The data was obtained from the largest and most comprehensive registry available in Australia, and our paper remains the first report of head and neck cancer trends in Australia, comparing First Nations peoples and non-Indigenous Australians. Please see the significant updates/improvements to the materials and methods section of the manuscript to reflect some of these challenges and the rationale for chosen statistical methodologies.
Comments 4: Survival rates: The Kaplan-Meier survival curves are appropriate for determining survival rates based on time-to-event data. This method effectively incorporates censored data to precisely calculate survival rates. The Cox proportional hazard model is suitable for estimating hazard ratios, facilitating comparisons of the risk of death between groups. Alternatively, the Binary logit model can be utilized for estimating 5-year survival rates.
Response 4: Thank you for your comment. We agree that Kaplan-Meier survival curves, Cox proportional hazard model or the Binary logit model are suitable for statistical analysis of survival data. However, the limitation we face is the data available in the ACD, which presents 1-year and 5-year relative survival rates as single percentages for the two time periods 1999-2006 and 2007-2014. Our study chose to utilise the 5-year relative survival rate. There is no person-level data or annual data available on head and neck cancer survivorship in the ACD. We again sought consultation from a statistician who viewed our data and advised us that survival data analysis is not possible. Hence, descriptive statistics would be the most suitable option to present the survival data. Please see the significant updates/improvements to the materials and methods section of the manuscript to reflect some of these challenges and the rationale for chosen statistical methodologies.
Comments 5: Time frame. The time trend data is very outdated, nearly 10 years old. I strongly urge the authors to use a current data set up to at least 2022 to reflect recent trends. Although the authors covered 33 years of data from 1982 to 2015, most of the analysis is based on data from 1998 onwards.
Response 5: Thank you for your comment. We agree that the time trend data is outdated. However, this data from 1998-2015 is the most current and comprehensive report of HNC data in Australia comparing First Nations peoples and non-Indigenous Australians. Despite the outdated data, our study remains the only research paper to comprehensively analyse HNC trends in Australia with a special focus on First Nations peoples. On a separate note, we have contacted the ACD to inquire when more recent reports of HNC will be publicly available, as we would welcome the opportunity for a follow-up study of this current research. At this stage, we are still awaiting a response. These limitations have been specified in the manuscript under materials and methods and the last paragraph of the discussion section.
Comments 6: Appendix and Table 1: Although it was mentioned in the statistical analysis section, I couldn’t find any justification for including Table 1 in the appendix when the data is not available.
Response 6: Thank you for pointing this out. We agree with this comment. Therefore, we have removed the Appendix section of the manuscript along with any references to the Appendix.
Comments 7: Figures. The figures do not follow the professional standards. Please follow the guidelines to ensure the scientific quality of the images.
Response 7: Thank you for pointing this out. We agree with your comment and apologise that the figures were not of scientific quality. Therefore, we have altered figures 1-7 to ensure: clear labels for each graph (e.g. A) Incidence in First Nations peoples), clear demarcations between graphs if they have been grouped under a single figure, removed the titles for each figure, ensured a legend for each graph even if it is grouped under a single figure, and overall provided higher resolution. Additionally, the fonts of the graphs’ labels and axes have been changed to Palatino Linotype to match the font of the manuscript. These changes can be seen in the manuscript under the results section.
|
||
|
4. Response to Comments on the Quality of English Language Nil.
|
||
|
5. Additional clarifications |
||
|
The title of the research paper has been changed to more clearly reflect the time data analysed, and this change has been reflected within the manuscript as well. Instead of 1982-2015, it has been changed to 1998-2015.
The materials and methods section of the manuscript has been heavily improved to ensure clarity to the readers.
References: reference number 25 was added to the reference list as an in-text reference. Reference number 21, 30, 31, 39, 40, 41, 47 and 48 was deleted from the reference list as an in-text reference.
|
||
Reviewer 2 Report
Comments and Suggestions for Authors
Review:
In this manuscript, the objective was to provide a comprehensive understanding of HNC incidence, mortality and survivorship among First Nations peoples, and draw comparisons with non-Indigenous Australians by retrospectively assessing trends with regards to age, sex, remoteness and geography from 1982-2015. This reviewer thought that this was an important research endeavour on an issue that is often under-researched. The discussion was well-written and conveyed many great points on the interpretation of their data.
1. On line 44, it would be more accurate if that last part of the sentence read as follows: “…, and infection with certain types of Human Papillomaviruses”
2. They Figure legends are quite short and could benefit from more information.
3. I think some sort of statistical test needs to be done on the data from Figure 7, to determine if there is any significance to what is being observed/reported.
4. The Tables in the appendix all have a column with “n.p”. Not sure what that means or if that is just an error. This should be fixed
Comments on the Quality of English LanguageMinor spelling and grammar issues.
Author Response
For research article
|
Response to Reviewer 2 Comments
|
||
|
1. Summary |
|
|
|
Thank you very much for taking the time to review this manuscript. Please find the detailed responses below and the corresponding revisions/corrections in track changes in the re-submitted files.
|
||
|
2. Questions for General Evaluation |
Reviewer’s Evaluation |
Response and Revisions |
|
Does the introduction provide sufficient background and include all relevant references? |
Yes |
Please see point-by-point response letter. |
|
Are all the cited references relevant to the research? |
Yes |
|
|
Is the research design appropriate? |
Can be improved |
|
|
Are the methods adequately described? |
Yes |
|
|
Are the results clearly presented? |
Can be improved |
|
|
Are the conclusions supported by the results? |
Yes |
|
|
3. Point-by-point response to Comments and Suggestions for Authors |
||
|
Comments 1: On line 44, it would be more accurate if that last part of the sentence read as follows: “…, and infection with certain types of Human Papillomaviruses”
Response 1: Thank you for pointing this out. We agree with this comment. Therefore, we have altered the last part of the sentence on page 2, line 44 in the manuscript to, “Recognised risk factors for HNC include tobacco consumption, excessive alcohol intake, and infection with certain types of human papillomavirus (HPV).”
Comments 2: The Figure legends are quite short and could benefit from more information.
Response 2: Thank you for pointing this out. We agree with this comment. Therefore, the legends for Figures 1-7 have been altered to include what the x- and y-axis denote in the graphWe hope this will benefit the reader in interpreting the individual graphs and overall figures. We have also significantly improved figures 1-7 to ensure: clear labels for each graph (e.g. A) Incidence in First Nations peoples), clear demarcations between graphs if they have been grouped under a single figure, removed the titles for each figure, ensured a legend for each graph even if it is grouped under a single figure, and overall provided higher resolution. Additionally, the fonts of the graphs’ labels and axes have been changed to Palatino Linotype to match the font of the manuscript. These changes can be seen in the manuscript under the results section.
Comments 3: I think some sort of statistical test needs to be done on the data from Figure 7, to determine if there is any significance to what is being observed/reported.
Response 3: Thank you for your comment. We agree that a statistical analysis of incidence and mortality rates by state distribution between First Nations peoples and non-Indigenous Australians would be a useful addition to our research. However, the limitation we face is the data available by the Australian Cancer Database (ACD). Categorised by state distribution, incidence data was available for 2009-2013, whereas mortality data was available for 2011-2015. Hence, no person-level or annual level data was available. We further sought consultation from a statistician who viewed our available data and advised us that no statistical tests could be performed, and descriptive statistics would be the most suitable option. Please also see the significant updates/improvements to the materials and methods section of the manuscript to reflect some of these challenges and the rationale for chosen statistical methodologies. It also needs to be noted that the data presented in this study is representative of the whole of Australian population, as there is no sampling variability which occurs when a sample is drawn from the whole population, analytical statistics would not offer any statistical inference.
Comments 4: The Tables in the appendix all have a column with “n.p”. Not sure what that means or if that is just an error. This should be fixed.
Response 4: Thank you for your comment. The n.p was used by the ACD to refer to ‘not publishable’ data because of small numbers, confidentiality or other concerns about the quality of the data. However, the appendix has been removed from the manuscript altogether, since there was no justification for including unavailable data.
|
||
|
4. Response to Comments on the Quality of English Language Thank you for your comment. We apologise that this was missed the first time around. Throughout the manuscript, changes have been made to correct spelling and grammatical issues.
|
||
|
5. Additional clarifications |
||
|
The title of the research paper has been changed to more clearly reflect the time data analysed, and this change has been reflected within the manuscript as well. Instead of 1982-2015, it has been changed to 1998-2015. |
||
The materials and methods section of the manuscript has been heavily improved to ensure clarity to the readers.
References: reference number 25 was added to the reference list as an in-text reference. Reference number 21, 30, 31, 39, 40, 41, 47 and 48 was deleted from the reference list as an in-text reference.
Reviewer 3 Report
Comments and Suggestions for Authors
The authors compared the health disparities in head and neck cancer between First Nations peoples and non-Indigenous Australians from 1998-2015 using data from the Australian Cancer Database. The results showed a significant disparity in regard to H/N cancer incidence and mortality. I have a few comments.
1. Line 30: “HNC encompasses ……affecting the sinuses, nose, oral cavity, salivary gland and throat”. I think this sentence is general, esp involving the throat. Would prefer more medical terms such as nasopharynx instead of nose, oropharynx, larynx et. instead of throat.
2. Line 50, “….. where men are at least twice likely to be diagnosed with and succumb to ….., a reflection of lifestyle-related choices”. Can the author be specific about what lifestyle-related choices? Also any other potential etiologies or risk factors?
3. Line 261, “solar radiation is … lip cancer”. Is this lip cancer more refers to one type of skin cancer?
4. Discussion part: The authors mentioned a lot regarding dental services. Just curious because in the US, most patients were firstly referred by family doctor to an ENT (ear, nose throat) doctor. Only a very small proportion of patients were referred by dentists. Were dental clinics the primary referring sources in Australia for H/N cancer patients?
Author Response
For research article
|
Response to Reviewer 3 Comments
|
||
|
1. Summary |
|
|
|
Thank you very much for taking the time to review this manuscript. Please find the detailed responses below and the corresponding revisions/corrections in track changes in the re-submitted files.
|
||
|
2. Questions for General Evaluation |
Reviewer’s Evaluation |
Response and Revisions |
|
Does the introduction provide sufficient background and include all relevant references? |
Yes |
Please see point-by-point response letter. |
|
Are all the cited references relevant to the research? |
Yes |
|
|
Is the research design appropriate? |
Yes |
|
|
Are the methods adequately described? |
Yes |
|
|
Are the results clearly presented? |
Yes |
|
|
Are the conclusions supported by the results? |
Yes |
|
|
3. Point-by-point response to Comments and Suggestions for Authors |
||
|
Comments 1:. Line 30: “HNC encompasses ……affecting the sinuses, nose, oral cavity, salivary gland and throat”. I think this sentence is general, esp involving the throat. Would prefer more medical terms such as nasopharynx instead of nose, oropharynx, larynx et. instead of throat. Response 1: Thank you for your comment. We agree with this comment, therefore, we have changed the sentence to incorporate medical terms such as nasopharynx, oropharynx and larynx replacing the use of ‘nose’ and ‘throat.’ |
||
|
Comments 2: Line 50, “….. where men are at least twice likely to be diagnosed with and succumb to ….., a reflection of lifestyle-related choices”. Can the author be specific about what lifestyle-related choices? Also any other potential etiologies or risk factors?
Response 2: Thank you for your comment. We agree with this comment, hence we have specified within that sentence that we are referring to modifiable risk factors more common in men such as tobacco and alcohol consumption. In the first paragraph of the introduction section, the last sentence also highlights common risk factors of head and neck cancer such as tobacco consumption, excessive alcohol intake, and the human papillomavirus.
Comments 3: Line 261, “solar radiation is … lip cancer”. Is this lip cancer more refers to one type of skin cancer? Response 3: Thank you for your comment. The use of the general term lip cancer was intentional, as it refers to malignant neoplasms of the external lip as well as skin of the lip. Comments 4: Discussion part: The authors mentioned a lot regarding dental services. Just curious because in the US, most patients were firstly referred by family doctor to an ENT (ear, nose throat) doctor. Only a very small proportion of patients were referred by dentists. Were dental clinics the primary referring sources in Australia for H/N cancer patients? Response 4: Thank you for your question. In Australia, general practitioners and ENT doctors also play a role in the diagnosis and referral process of HNC, however, multiple studies focusing on Australia have highlighted the critical role of dental professionals in initial detection and management of HNC. Furthermore, the AIHW singularly mentions dentists as having a role in oral cancer detection.
|
||
|
4. Response to Comments on the Quality of English Language Nil.
|
||
|
5. Additional clarifications |
||
References: number 23 has been replaced with a new journal article.
Round 2
Reviewer 1 Report
Comments and Suggestions for Authors
The authors have revised the article as suggested and addressed my questions and comments. However, I am following up with the authors regarding the following queries to further improve the manuscript.
Cancer incidence and mortality rates can fluctuate significantly across different years and population characteristics. Applying a log transformation can stabilize the variance, making the data more homoscedastic, which is essential for precise model fitting and reliable statistical inference. Although this was not mentioned in the statistical analysis section, it is worth checking if the log scale was used during model building.
While joint point regression does not reveal any insights into cancer rates or survivals for First Nations People, I request the authors to discuss the APC from the joint points and provide any recommendations for future HNC incidence and mortality rates in the Non-Indigenous cohort in the discussion section.
Please double-check in the title whether it should be "First Nations peoples" or "First Nations People."
Author Response
For research article
|
Response to Reviewer 1 Comments
|
||
|
1. Summary |
|
|
|
Thank you very much for taking the time to review this manuscript. Please find the detailed responses below and the corresponding revisions/corrections in track changes in the re-submitted files.
|
||
|
2. Questions for General Evaluation |
Reviewer’s Evaluation |
Response and Revisions |
|
Does the introduction provide sufficient background and include all relevant references? |
Yes |
Please see point-by-point response letter. |
|
Are all the cited references relevant to the research? |
Yes |
|
|
Is the research design appropriate? |
Yes |
|
|
Are the methods adequately described? |
Can be improved |
|
|
Are the results clearly presented? |
Can be improved |
|
|
Are the conclusions supported by the results? |
Can be improved |
|
|
3. Point-by-point response to Comments and Suggestions for Authors |
||
|
Comments 1: Cancer incidence and mortality rates can fluctuate significantly across different years and population characteristics. Applying a log transformation can stabilize the variance, making the data more homoscedastic, which is essential for precise model fitting and reliable statistical inference. Although this was not mentioned in the statistical analysis section, it is worth checking if the log scale was used during model building. Response 1: Thank you for your comment. The Joinpoint regression analysis performed in our study utilises a log transformation when calculating annual percent changes (APC). This approach makes the date more homoscedastic, as you pointed out. The first step involved in estimating the APC is log transformation of original data points following by segment fitting, joinpoints and APC calculation. We have now added a sentence (see below) in the methods section to provide information to readers on the log transformation. “Joinpoint regression uses log transformation of the incidence and mortality data resulting in a linear model fitted to the trend on a logarithmic scale facilitating the direct comparison of relative differences and annual percent changes overs a specific period in the output.” Comments 2: While joint point regression does not reveal any insights into cancer rates or survivals for First Nations People, I request the authors to discuss the APC from the joint points and provide any recommendations for future HNC incidence and mortality rates in the non-Indigenous cohort in the discussion section. Response 2: Thank you for your comment. As per your request, we have discussed in the second paragraph of the discussion section the APC’s relating to the non-Indigenous cohort and made recommendations for future incidence and mortality rates. Please see the paragraph below. “On the other hand, there were statistically significant declines in incidence and mortality in the non-Indigenous cohort. More specifically, there was a decrease in incidence in non-Indigenous males with an APC of -2.68 (p<0.05) between 1998-2003, and -1.16 (p<0.05) between 2006-2013, and a decline in mortality in non-Indigenous females between 1998-2015 with an APC of -1.46 (p<0.05). At the population level, there was a decline in mortality of non-Indigenous Australians between 2001-2006 with an APC of -4.55 (p<0.05). These significant declines are encouraging findings in the non-Indigenous cohort; however, more targeted interventions are required to produce such results consistently at the population-level.” Comments 3: Please double-check in the title whether it should be "First Nations peoples" or "First Nations People." Response 3: Thank you for your comment. We have changed it to First Nations people, taking into consideration the recently recommended naming convention for Aboriginal and Torres Strait Islander peoples.
|
||
|
4. Response to Comments on the Quality of English Language Nil.
|
||
|
5. Additional clarifications |
||
References: number 23 has been replaced with a new journal article.
Reviewer 2 Report
Comments and Suggestions for Authors
The authors have addressed all my comments. Thank you.
Author Response
Thank you.